# Effect of Dielectric Barrier Discharge Plasma against *Listeria monocytogenes* Mixed-Culture Biofilms on Food-Contact Surfaces

**DOI:** 10.3390/antibiotics12030609

**Published:** 2023-03-19

**Authors:** Min Gyu Song, Pantu Kumar Roy, Eun Bi Jeon, So Hee Kim, Min Soo Heu, Jung-Suck Lee, Jae-Suk Choi, Jin-Soo Kim, Shin Young Park

**Affiliations:** 1Institute of Marine Industry, Department of Seafood Science and Technology, Gyeongsang National University, Tongyeong 53064, Republic of Korea; 2Research Center for Industrial Development of Seafood, Gyeongsang National University, Tongyeong 53064, Republic of Korea; 3Department of Food and Nutrition, Gyeongsang National University, Jinju 52828, Republic of Korea

**Keywords:** DBD plasma, non-thermal-plasma, biofilm, *Listeria monocytogenes*, food-contact surfaces, inactivation

## Abstract

*Listeria monocytogenes* is a major foodborne pathogen. Various methods can be used to control biofilms formed by foodborne pathogens. Recently, the food industry has become interested in plasma, which can be used as a non-thermal technology with minimum changes to product quality. In this study, the effects of dielectric barrier discharge (DBD) plasma on *L. monocytogenes* mixed-culture biofilms formed on stainless steel (SS), latex hand glove (HG), and silicone rubber (SR) were investigated. DBD plasma effectuated reductions of 0.11–1.14, 0.28–1.27 and 0.37–1.55 log CFU/cm^2^, respectively. Field emission scanning electron microscopy (FE-SEM) demonstrated that DBD plasma cuts off intercellular contact and induces cell decomposition to prevent the development of biological membranes. It was confirmed that the formed biofilms collapsed and separated into individual bacteria. Our findings suggest that DBD plasma can be used as an alternative non-heating sterilization technology in the food industry to reduce biofilm formation on bacterial targets.

## 1. Introduction

A common food-borne pathogen *Listeria monocytogenes* can result in a deadly illness called listeriosis in 20–30% of vulnerable people [1]. The presence *of L. monocytogenes* on food-contact surfaces can contaminate food and infect a human upon ingestion [2]. Such contaminated foods cause foodborne infections, which represents a severe public health concern [3]. Several mechanisms for pathogen survival and colonization, such as the formation of biofilms and motility, are connected to pathogen infections [4].

Biofilms are three-dimensional structures composed of microorganisms that grow on a solid surface as membranes in a self-dispersed polymeric substrate [5,6]. Reducing the contamination and illnesses caused by *L. monocytogenes* is a major goal for biofilm research in the food industry [3]. *L. monocytogenes* biofilms are found on various surfaces, such as natural aquatic, drinking water, biological tissue, medical devices, and food industry sites [7]; they are 10–1000 times more resistant to removal from the surface than the planktonic bacteria and cause food poisoning [3,8]. *L. monocytogenes* is able to produce biofilms on both food and food-contact surfaces, which helps it survive in conditions where food is being processed [1,4]. Various technologies such as UV-C and chlorine have been used to prevent contamination of foodborne pathogens; these new technologies are linked to green solutions.

A different approach to controlling *L. monocytogenes* is to treat the plasma with dielectric barrier discharge (DBD). Irving Langmuir invented the word plasma for the first time in 1928 [9]. Plasma is the fourth state of matter after solid, liquid, and gas according to the internal energy sizes of the particles constituting the material. When the gas has higher energy, it is separated into ions and electrons, whose energies are parallel to each other [10]. The plasma can be classified as thermal plasma and non-thermal plasma. Thermal plasma operates at very high pressures and requires extensive power for production, whereas non-thermal plasma is produced at low pressure, atmospheric temperatures, or ambient temperatures and requires less energy for generation [11]. Non-thermal plasma can be used for the reduction of contamination of food without worrying about changes in food quality or nutritional value [12]. The specific component plasma includes active radicals with high chemical reactivity, which can be inactivated microorganisms, and is divided into reduced-pressure plasma and atmospheric-pressure plasma. Reduced-pressure plasma has advantages in that it is easy to control with respect to the generation speed; uniform plasma may be generated, but a facility to lower pressure to a state close to the vacuum is required and continuous treatment is difficult [13]. In contrast, atmospheric-pressure plasma can be used in the food industry owing to the simple equipment required, continuous processing, and low equipment costs. The discharge forms of atmospheric-pressure plasma include DBD, corona discharge, microwave discharge, and arc discharge [14]. Among them, DBD operates under extensive non-equilibrium conditions. Its discharge form is suitable for food treatment because it can discharge high power, has no electrical impact, and allows treatment of large areas [15].

*L. monocytogenes* is commonly found in seafood, marine products, and water [16]. However, *L. monocytogenes* is not limited to seafood and has been detected in various foods. Cross-contamination of food and food-contact surfaces has been reported in food industries, such as cheese, ice cream, pork, minced meat, white fish, salmon, and blue crabs, and due to these characteristics, the food industry should continuously aim to identify a sterilization method capable of removing *L. monocytogenes* biofilms [2,17,18,19].

The removal of *L. monocytogenes* biofilms using plasma has been demonstrated [20,21,22,23]. However, research on biofilm reduction on food-contact surfaces against *L. monocytogenes* biofilms—a major cause of cross-contamination in the food industry—using DBD plasma, conducted experiments and demonstrated that the technology works in different situations. Therefore, in the current study, we investigate the reduction effects of DBD plasma after *L. monocytogenes* biofilm formation on stainless steel (SS), hand gloves (HG), and silicon rubber (SR)—surfaces that are mainly used in food processing.

## 2. Results

### 2.1. Reduction Effects of DBD Plasma against the L. monocytogenes Mixed-Culture Biofilm on SS Surface

The reduction effects of DBD plasma on *L. monocytogenes* mixed-culture biofilms were evaluated by measuring biofilms on three distinct food-contact surfaces 5–60 min after treatment. *L. monocytogenes* mixed-culture biofilms on the SS surface are shown in Figure 1. As the processing time increased, the total *L. monocytogenes* mixed-culture biofilms decreased significantly after DBD treatment (*p* < 0.05). After treatment for 5, 15, 30, 45, and 60 min, the overall mean of *L. monocytogenes* mixed culture was 4.97 (0.11 log reduction), 4.71 (0.37 log reduction), 4.43 (0.65 log reduction), 4.04 (1.04 log reduction), and 3.94 (1.14 log reduction) log CFU/cm^2^, respectively, compared with the initial titer of the *L. monocytogenes* mixed-culture biofilms, i.e., 5.08 log CFU/cm^2^. Treatment for 60 min showed a significant difference (*p* < 0.05) compared with other groups except the 45 min-treated group. The survival of *L. monocytogenes* mixed-culture biofilms on the SS surface in response to DBD plasma treatment was calculated using a first-order kinetic model (Figure 1). Table 1 lists the values for the first-order kinetic model’s parameters. Based on a first-order kinetic model, the D-values for *L. monocytogenes* mixed-culture biofilms on the SS surface were determined to be 50.00 min, with an R^2^ of 0.98.

### 2.2. Reduction Effects of DBD Plasma against L. monocytogenes Mixed-Culture Biofilms on HG Surface

Overall *L. monocytogenes* mixed-culture biofilms reduction reduced significantly (*p* < 0.05) with increasing time after 60 min treatment of DBD plasma (Figure 2). For DBD plasma treated for 5, 15, 30, 45, and 60 min, the average of the *L. monocytogenes* mixed-culture biofilm was reduced to 5.06 (0.28 log reduction), 4.70 (0.64 log reduction), 4.41 (0.93 log), 4.21 log (1.27 log) and 4.27 log, respectively, compared with the initial titer of the *L. monocytogenes* biofilm 5.34 log CFU/cm^2^. There was a significant difference in the treatment for 60 min when compared with other treated and control groups except the 45 min-treated group of DBD plasma. The survival of mixed culture *L. monocytogenes* biofilms on the HG surface in response to DBD plasma treatment was calculated using the first-order kinetic model (Figure 2). The parameters of the first-order kinetic model are presented in Table 1. Using a first-order kinetic model, we determined a 50.25 min D-value and an R^2^ of 0.92 for the *L. monocytogenes* mixed-culture biofilm on the HG surface.

### 2.3. Reduction Effect of DBD Plasma against L. Monocytogenes Mixed-Culture Biofilm on SR Surface

*L. monocytogenes* mixed-culture biofilms on the SR surface are shown in Figure 3. As the processing time increased, the total *L. monocytogenes* mixed-culture biofilm decreased significantly after DBD treatment (*p* < 0.05). After treatment for 5, 15, 30, 45, and 60 min, the overall mean of *L. monocytogenes* mixed-culture biofilms was 5.19 (0.37 log reduction), 4.98 (0.58 log reduction), 4.30 (1.26 log reduction), 4.21 (1.35 log reduction), and 4.01 (1.55 log reduction) log CFU/cm^2^, respectively, compared with the initial titer of *L. monocytogenes* mixed-culture biofilms, i.e., 5.56 log CFU/cm^2^. Significant difference was found in the 60 min-treated group compared with other groups. The survival of *L. monocytogenes* mixed-culture biofilms on the SR surface in response to DBD plasma treatment was calculated using a first-order kinetic model (Figure 3). First-order kinetic model calculations of the *L. monocytogenes* mixed-culture biofilm on the SR surface calculated a D-value of 39.53 min and an R^2^ of 0.92 (Table 1).

### 2.4. Visual Confirmation of Biofilm Reduction by DBD Plasma Using FE-SEM

The inhibitory effects of DBD plasma on *L. monocytogenes* mixed-culture biofilms grown on SS, HG and SR surfaces were visually confirmed by FE-SEM, and the results are shown in Figure 4. The non-DBD plasma-treated samples, i.e., on the surfaces of SS (Figure 4A), HG (Figure 4C) and SR (Figure 4E), exhibited intact biofilms connected with extracellular polymeric substance (EPS). In samples of DBD plasma treated for 60 min, the surface of SS (Figure 4B), HG (Figure 4D) and SR (Figure 4F) did not show the formation of an intact biofilm, and the bacterial cells treated with DBD appeared rough and irregular, indicating that the cells had lost their regular structure. Compared to the control group, only some bacteria that existed as individual cells remained. Thus, disintegration of biofilm formation was significant compared with the control (Figure 4).

### 2.5. Comparison of D-Value

When comparing the D-values with respect to the DBD plasma treatment time, there was no significant difference in the biofilms remaining on SS and HG; however, for *L. monocytogenes* in biofilms on SR, inactivation times were shorter. The D-values of SS, HG, and SR were 50.00, 50.25 and 39.53 min, respectively, and treatment with DBD plasma in SR was found to be the most effective compared to that on other surfaces. It was revealed that surface qualities influenced these results. Plasma is a unique gas with electrical properties. Due to the great electrical conductivity of SS, plasma may have circulated across the material when its surface was treated. Therefore, the efficiency was lower in the case of SS than in the SR. In the case of HG, as shown in Figure 4, the surface is not smooth, and many gaps exist. It was determined that the efficiency of plasma was lower in this case than that observed in the case of the relatively flat SR because the biofilms formed in the gap were difficult to remove.

## 3. Discussion

The DBD plasma method creates a uniform plasma, and the circuit configuration is relatively simple [24]. When plasma is generated, ions and electrons are separated and only charged particles are formed. Additionally, active radicals with high chemical reactivity, hydrogen peroxide (H_2_O_2_), and active species, such as ozone (O_3_), are generated. According to reports, these active species are effective at sterilizing food-poisoning bacteria by penetrating their cells and damaging the lipids and proteins in their cell membranes [15]. This study found that DBD plasma effectively reduced biofilms on different surfaces. As the treatment time with DBD plasma increased, a decrease was observed in all strains, and it was found that DBD plasma was effective against foodborne pathogens. We could show that with increasing exposure time, the number of viable bacteria recovered from biofilms decreased. When the *L. monocytogenes* mixed-culture biofilms on the SS, HG, and SR surfaces were treated for 60 min, they reduced to 1.14, 1.27 and 1.55 log CFU/cm^2^, respectively (Figure 1, Figure 2 and Figure 3). Previous studies reported that ROS and/or RNS are the active compounds that kill bacteria via DBD, while UV irradiation and changes in temperature are either small factors or have no effect [25]. Moreover, various materials treated by plasma can attain antimicrobial activity [26]. Kim et al. [27] inoculated *V. parahaemolyticus* on a wooden chopping board—a food-contact surface—and treated it with DBD plasma for 60 min. Govaert et al. [28] reported that treatment with DBD plasma for 30 min reduced *L. monocytogenes* and *Salmonella* Typhimurium biofilms by 2.40 and 2.95 log CFU/g, respectively, and Cuiet et al. [29] effectively deactivated the biofilms of *L. monocytogenes* using cold nitrogen plasma. Modic et al. [30] reported that *S. aureus* biofilms were inactivated by cold atrophic pressure plasma. In addition, cold atmospheric plasma effectively removes biofilms formed by various bacteria such as *Pseudomonas aeruginosa*, *P. fluorescens*, and *S. epidermidis* [31]. DBD plasma at 5, 10, and 15 min with 2 kHz decreased by 2.84 log (~99.9%), 2.88 log (~99.9%), and 2.89 log (~99.9), respectively, against *P. aeruginosa* [32]. Plasma has shown effects on both Gram-negative and Gram-positive bacteria. Scholtz et al. [33] reported that non-thermal plasma could remove biofilms formed by Gram-negative and Gram-positive bacteria in 10 min, and in a study by Mai-Prochnow et al. [34], plasma was found to be effective against both Gram-negative and Gram-positive bacteria. Plasma was applied to the biofoulant (activated sludge) on the membrane surfaces for 3–10 min, and the appropriate dosage was found to be 13–42 J cm^2^. The procedure for removing biofouling is quite effective, depending on the type of membrane and the amount of time it was exposed to non-thermal plasma [35]. Depending on treatment time, biofilm was reduced up to 3.5 log CFU/cm^2^ against *L. monocytogenes* and *S.* Typhimurium [36]. It has been reported that planktonic bacteria were completely inactivated within 5 min of plasma treatment, whereas bacteria grown as biofilms showed more than a 3 log reduction (99.9%) at the end of 15 min against P. aeruginosa [32].

DBD plasma inhibits and eradicates biofilms produced by foodborne bacteria. Bacteria lose their normal structure in response to plasma-mediated DBD due to interrupted cell-to-cell interactions. The growth of well-organized biofilms and colonization are both facilitated by these intercellular interactions [3]. When these interactions are severed, the biofilm cells become split and are simply eliminated by washing. FE-SEM images of the mixed culture of *L. monocytogenes* showed that DBD plasma broke down the cell-to-cell interactions (Figure 4B,D,F).

The *L. monocytogenes* mixed-culture biofilms on the SS, HG and SR surfaces were visually confirmed by FE-SEM, and the results are shown in Figure 4. Bacteria were accumulated on the surfaces of the SS, HG and SR coupons that were not treated with DBD plasma (Figure 4A,C,E) to generate EPS, confirming that the intact biofilm was clustered. However, in the case of DBD plasma-treated samples, most bacteria became loose and biofilm formation was no longer observed. This suggests that DBD plasma effectively inhibits biofilm formation. In a study by Flynn et al. [37], when *Acinetobacter baumannii* biofilms were treated with cold plasma, scanning electron microscopy (SEM) confirmed that the cell state of the intact biofilm was lost and changed to a formless state. In addition, Cui et al. [38] reported *E. coli* O157 biofilm formation in lettuce, and upon the treatment of this biofilm with cold nitrogen plasma, SEM revealed severely damaged cell membranes in the biofilm. Similar to a different study, regular DBD therapy at both frequencies revealed disturbed biofilms and cells that seemed to have structural damage [38], and the reason for this may be the use of gas in the plasma, which increases the production of chemically reactive species, thereby increasing the antimicrobial effect. Previously, Lee et al. [39] reported that FE-SEM analysis of atmospheric pressure plasma jet-treated biofilms of Gram-positive and Gram-negative bacteria on a titanium surface revealed decomposed cell membranes during treatment and severe damage after treatment [40]. Several studies have demonstrated the effectiveness of biofilm reduction using UV-C, a sterilization technique similar to DBD plasma [41]. UV-C irradiation emits short wavelengths (180–280 nm) of light and has a high energy concentration; therefore, it has a robust antibacterial effect [41]. Much research has noted that DBD plasma has deep and extensive penetration; however, there are not enough investigations on the various surfaces that come into contact with food. In order to find a general physical antibacterial alternative that is safe, environmentally friendly, and capable of reducing pathogenicity, this study looked at the antimicrobial effect of DBD plasma.

## 4. Materials and Methods

### 4.1. Preparation of Food-Contact Surfaces

The surfaces were prepared in accordance with the methodology proposed in previous studies [42,43,44]. Briefly, stainless steel (SS), latex hand glove (HG), and silicon rubber (SR) used in the experiments were 2 × 2 cm^2^ in size. The area available for bacterial cell adhesion is increased/decreased by the roughness/smoothness of a surface. Surface imperfections facilitate initial colonization because they shield adherent germs from external stimuli and encourage transition from reversible to permanent attachment. The SS, HG, and SR coupons were soaked in 70% ethanol, dried on each side sufficiently on a clean bench (15 min) to prevent natural contamination.

### 4.2. Preparation of Strains and Culture Conditions

In this study, three strains of *L. monocytogenes*, ATCC 15313, ATCC 19115, and ATCC 1917 were used for mixed-culture biofilm formation. The strains were stored in trypticase soy broth (TSB, Difco Laboratories, Detroit, MI, USA) containing 30% glycerol in a deep freezer at −80 ℃. *L. monocytogenes* was grown for 24 h in 10 mL of TSB at 30 ℃. For bacterial activation, this procedure was carried out twice. At least twice with phosphate-buffered saline (PBS, Oxoid, UK, Hampshire), the suspension was centrifuged for 10 min at 4695× *g* at 4 °C. Equal volumes of each strain were mixed to create a cocktail culture. Then, 0.1% of buffered peptone water (PW, Oxoid, Basingstoke, England) was used to dilute serially to make the final bacterial solution to 10^5^ log CFU/mL of the mixed suspension for further studies [45,46,47].

### 4.3. Biofilm Formation

Biofilm was prepared in accordance with previous studies with slight modifications [47]. Sterilized SS, HG, and SR pieces were placed in a 50 mL Falcon tube containing 10 mL TSB with 100 µL (10^5^ log CFU/mL) of bacterial suspension and incubated for 24 h at 30 °C.

### 4.4. DBD Plasma Treatment

After the formation of biofilms on the SS, HG, and SR surfaces, DBD plasma (μ-DBD Surface Plasma Generator, Model; Micro DBD plasma) treatment was initiated (Figure 5). The experiment was conducted in triplicate. The equipment used for the DBD plasma treatment was a plasma generator (Plasma Biomedical Research Institute in Seoul, Republic of Korea). A 1.5 L/min nitrogen flow was used to generate DBD plasma between a glass and metal mesh grid on the back glass. The DBD plasma transferred a low discharge voltage of approximately 1 kV and discharge peak current of 40 mA at a driving frequency of 43 kHz [48]. During the treatment, the distance between the plasma-emitting electrode and the samples was maintained at 3 mm. The SS, HG, and SR surfaces were exposed to the DBD plasma for 5, 10, 15, 30, 45, and 60 min, respectively.

### 4.5. Detachment of Biofilms

The procedure was performed as a slight modification of a previously described protocol [3,44]. After biofilms formation, the surfaces were simply washed with distilled water to remove the loosely adhering cells. Briefly, to decompose the biofilm on the washed SS, HG, and SR surfaces, the surfaces were immersed in 10 mL PW containing 10 sterile glass beads and vortexed for 2 min. The removed cells were vortexed and diluted in PW for enumeration, and bacterial suspensions of *L. monocytogenes* mixed cultures were inoculated and diffused into PALCAM agar (Oxoid, UK, Hampshire). The number of colonies on the plates was counted after 24 h incubation at 30 °C, and biofilm cells were counted as log CFU/cm^2^.

### 4.6. Determination of D-Values of DBD Plasma

A first-order kinetics model was used to calculate the D-values for the reduction of *L. monocytogenes* mixed cultures on the SS, HG, and SR coupons. The D-value represents the plasma treatment time necessary to achieve a reduction in bacterial density of 1 log CFU/cm^2^ (>90%). A regression line that accurately described the data points had an R^2^ value near to 1. Using the formula D = −1/slope from a microbial population vs. DBD plasma curve, the D-value (min) was determined.

### 4.7. Field Emission Scanning Electron Microscopy (FE-SEM)

The surfaces were observed using FE-SEM to confirm the biofilm inhibition by DBD plasma on food-contact surfaces (SS, HG and SR). Samples were prepared in accordance with the protocol proposed in previous studies [43,49]. For 4 h, the samples were fixed in 2.5% glutaraldehyde made in PBS. After that, they had 15 min of treatment with gradient ethanol (50, 60, 70, 80 and 90%, respectively). For dehydration, the materials were submerged in (33, 50, 66, and 100% hexamethyldisilazane made in ethanol). The samples were then used for FE-SEM (Hitachi/Baltec, Tokyo, Japan) after being platinum sputter coated (Q150T Plus, Quorum, UK) and dried in a fume chamber for 3 h.

### 4.8. Statistical Analysis

Each experiment was carried out for statistical analysis at least three times. All data were presented as a mean with standard error of the mean (SEM). The SPSS software (version 12.0; SPSS Inc., Chicago, IL, USA) statistical program applied one-way ANOVA and Duncan’s multi-range test to confirm significant differences at the 5% level (*p* < 0.05).

## 5. Conclusions

Our findings demonstrate the potential antibacterial efficacy of DBD plasma in biofilm prevention on different food-contact surfaces. DBD plasma significantly reduced the number of viable bacterial cells on SS, HG, and SR (1.14, 1.27 and 1.55 log CFU/cm^2^, respectively), and damaged intercellular connections (FE-SEM images). Our results reveal that plasma treatment disrupts the integrity of the biofilms on food-contact material. Therefore, DBD plasma can be developed as an alternative method for removing biofilms of foodborne pathogens and lowering the risk of foodborne diseases.

## Figures and Tables

**Figure 1 antibiotics-12-00609-f001:**
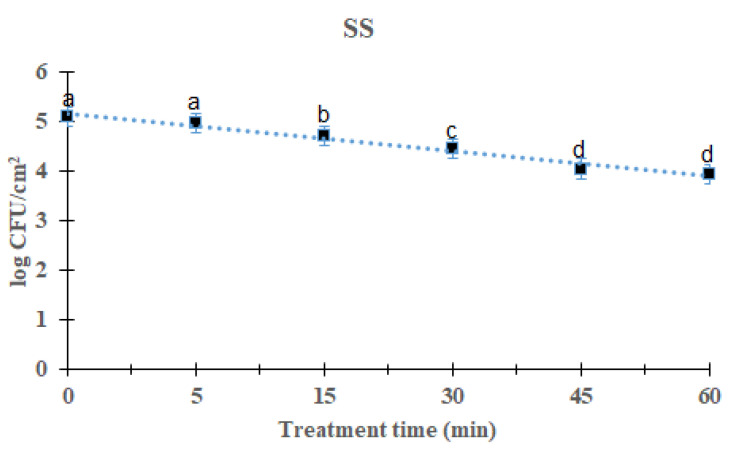
Survival curves of *Listeria monocytogenes* biofilm in stainless steel treated by DBD plasma using the first-order kinetic model. The data were represented by the mean ± standard deviation of the three independent replicates. ^a–d^ Values marked with different letters within each treatment are significantly different by Duncan’s multiple range test (*p* < 0.05).

**Figure 2 antibiotics-12-00609-f002:**
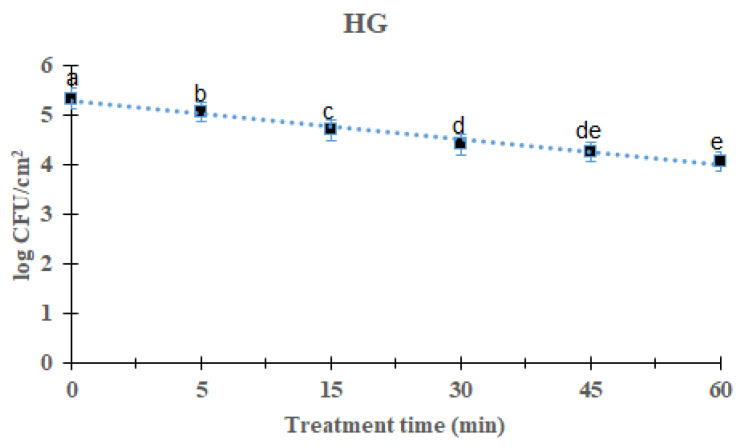
Survival curves of *Listeria monocytogenes* biofilm in a latex hand glove treated by DBD plasma using the first-order kinetic model. The data were represented by the mean ± standard deviation of the three independent replicates. ^a–e^ Values marked with different letters within each treatment are significantly different according to Duncan’s multiple range test (*p* < 0.05).

**Figure 3 antibiotics-12-00609-f003:**
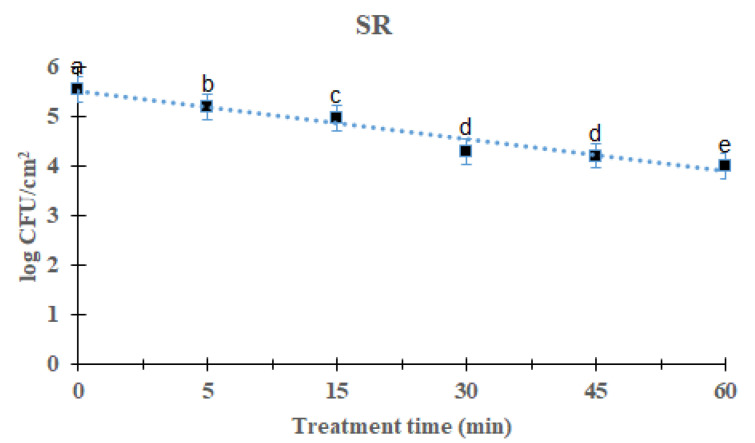
Survival curves of *Listeria monocytogenes* biofilm in silicon rubber treated by DBD plasma using the first-order kinetic model. The data were represented by the mean ± standard deviation of the three independent replicates. ^a–e^ Values marked with different letters within each treatment are significantly different by Duncan’s multiple range test (*p* < 0.05).

**Figure 4 antibiotics-12-00609-f004:**
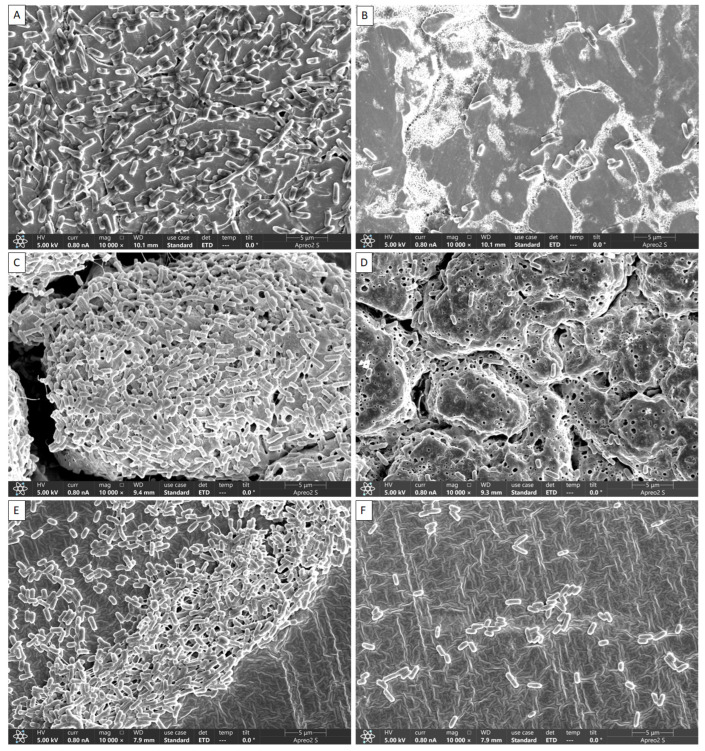
Representative scanning electron micrographs of *Listeria monocytogenes* mixed-culture biofilms formation DBD plasma treatment on the SS surfaces: (**A**) control; (**B**) 60 min DBD plasma treatment, HG surfaces: (**C**) control; (**D**) 60 min DBD plasma treatment and SR surfaces: (**E**) control; (**F**) 60 min DBD plasma treatment.

**Figure 5 antibiotics-12-00609-f005:**
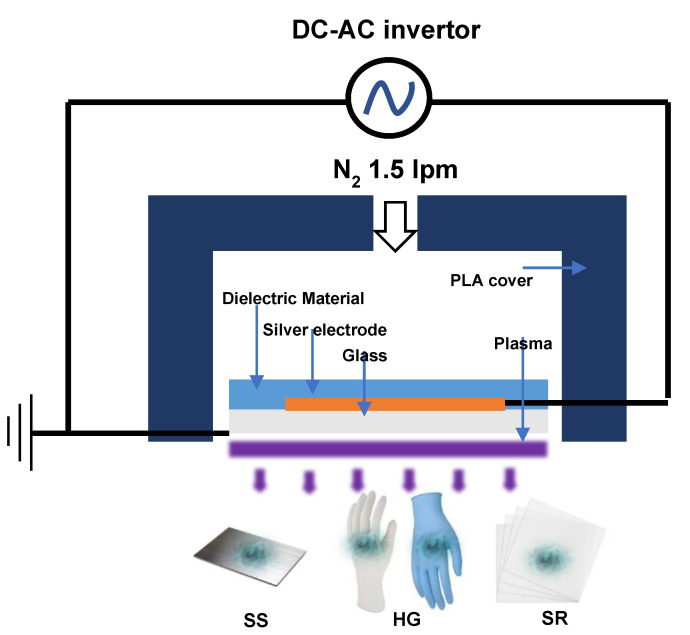
Illustration of the DBD plasma used in our study. Direct current (DC), alternating current (AC), Polylactic acid (PLA).

**Table 1 antibiotics-12-00609-t001:** Effect of DBD plasma treatment on D-values of *L. monocytogenes* mixed-culture biofilm, and reduction by first-order kinetics model in the SS, HG and SR.

	D-Value (min)	R^2^	y = ax + b
SS	50.00 ± 2.14 ^a^	0.98	y = −0.020x + 5.0438
HG	50.25 ± 9.36 ^a^	0.92	y = −0.0199x + 5.1545
SR	39.53 ± 1.88 ^b^	0.92	y = −0.0253x + 5.3618

The data present means for three samples with standard deviations (three samples/treatment). D-values, decimal of log reduction time. R^2^, correlation coefficient. ^a,b^ Values marked with different letters within each treatment are significantly different by Duncan’s multiple range test (*p* < 0.05).

## Data Availability

Data are contained within the article.

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
