# Peer review of "Effect of Dielectric Barrier Discharge Plasma against Listeria monocytogenes Mixed-Culture Biofilms on Food-Contact Surfaces"

_antibiotics, 2023, doi:10.3390/antibiotics12030609_

Round 1

Reviewer 1 Report

The study provides interesting insights into the alternative green solution for sterilization treatment of food contact surfaces against Listeria monocytogenes in the food industry.

Line 18: could the results be expressed as a percentage?

Lines 38-43: reduce the use of the word biofilm

Line 44: add "in the food industry" after "for biofilm research"

Line 45: emphasize that new technologies are linked to green solutions

Line 102: perhaps use "pieces" instead of "coupons"

Line 107: how many replicates were per treatment?

Line 160-163: could results be expressed as percentages?

Lines 252-255: don't repeat results, just discuss them

Reviewer 2 Report

The manuscript reports the reduction of viable Listeriae in biofilms when exposed to DBD plasma (with nitrogen gas).

General: the authors often use the term "sterilization", but they present reductions in numbers of bacteria, and no sterilization.

There are some issues:

line 14: suggest: which can be used as a non-thermal sterilization technology with minimum changes to product quality.

line 17: suggest: DBD plasma effectuated reductions of 0.11-1.14..., respectively.

line 25: suggest: "non-thermal-plasma"; suggest to replace "sterilization" by "inactivation" or similar word

line 19: superscript "2" in cm2.

line 31: delete "and infect it upon ingestion."

lines 34-43: the sentences could be rearranged.  I suggest to move the sentence "L. monocytogenes can form biofilms..." down after "Biofilms are difficult to remove.... [3, 8]."

line 49: plasma ist not the fourth substance, it is the fourth state of matter.

line 52: replace "was separated using" by "can be classified as.."

line 56: sterilization of food by plasma is usually not possible, but a reduction in the order of xx log cycles

line 58: microorganisms are not sterilized, but inactivated, killed...

line 68: "can handle a large part" replace by "and allows treatment or large areas"

line 87: suggest to delete "and used"

line 85: some details on surface smoothness/roughness of stainless steel should be given.

line 91: "tryptic soy broth" or "trypticase soy both"

lines 92, 96: after the acronym either "," or ";", but consistently.

line 98: density adjustment should be explained.

line 107: some information on the prevailing plasma gas species in the device should be given.

line 123: "adhering"

line 163: in Fig. 2, it seems that 60 min treatment was sign. diff. from 0, 5,15,30 but not from 45 min. This should be explained better. Also, in Fig. 2 the meaning of the  lettering must be explained. (Similar for line 185, Fig.3; line 208, Fig.4). 

line 169: "2" superscript in "R2" 

line 224: it was not inhibition, since the biofilm has been formed before plasma exposure; it was disintegration.

line 233: "the biofilms on SR..." maybe better: however, for Listeria in biofilms on SR, inactivation times were shorter"

line 252: suggest "We could show that with increasing exposure time, the number of viable bacteria recovered from biofilms decreased."

line 255: please check: "decreased to" means that 1.14 etc. were the numbers of remaining bacteria, but I think the authors report the reduction, not the survivors.

line 257: reword "is the main thing" by e.g. "are the active compounds that inactivate bacteria"

line 279: suggest "inactivated" instead of "decontaminated"

line 303: after "Previously", replace "." by ","

line 324: "implant materials" ? it should be "food contact material" or so.

line 325: the sentence must be corrected: "... was attributed to.." better to end sentence "... materials." and then start next sentence. However, the authors provide no proof in this study that RON or RNS were the causative agents, they conclude that from other references.

Reviewer 3 Report

The authors report on an experiment conducted to observe the effect of dielectric discharge plasma on Listeria monocytogenes biofilms grown on materials commonly encountered as contact surfaces in the food industry.

The content of the manuscript was consistent with the title of the manuscript.

The sections into which Materials and Methods, as well as the Results, are separated, increase the readability of the manuscript.

Terms are adequately explain in the text of the manuscript.

Detailed comments

Line 31 Change "infect it" to "infect a human"

Line 37 Delete first sentence

Line 41 - 42  The sentence "Biofilms are 10-1000 times more resistant than the planktonic bacteria" is incomplete. To which type of resistance are the authors referring? Consider combining with the first part of the next sentence if you are referring to the difficulty to remove the biofilm from a surface. "cause food poisoning" will then be integrated with a re-write of the sentence beginning "Reducing the contamination ..."

Line 45 and 76 to77  The phrases "but new technologies must be continuously developed" and "using DBD plasma should be continuously conducte" beg the question "Why".  If you have conducted experiments and demonstrated that the technology works in a particular situation, why continuously conduct the same experiments. Maybe you mean to say that the experiments should be done to determine if it works in different situations encountered, for example, the Roy et al papers referenced report on the use of DBD on different pathogens. Please modify your sentences accordingly.

Line 73 Remove "study".

Line 314 "microbial antimicrobial of foods"
